# Canoeing Motion Tracking and Analysis via Multi-Sensors Fusion

**DOI:** 10.3390/s20072110

**Published:** 2020-04-08

**Authors:** Long Liu, Sen Qiu, ZheLong Wang, Jie Li, JiaXin Wang

**Affiliations:** 1The Laboratory of Intelligent System, Dalian University of Technology, Dalian 116024, China; qiu@dlut.edu.cn (S.Q.); wangzl@dlut.edu.cn (Z.W.); 1165530693@mail.dlut.edu.cn (J.L.); wangjx19890828@mail.dlut.edu.cn (J.W.); 2Department of Electrical & Information Engineering, Dalian Neusoft Institute of Information, Dalian 116023, China

**Keywords:** rowing sport, motion reconstruction, inertial sensor, data fusion

## Abstract

Coaches and athletes are constantly seeking novel training methodologies in an attempt to improve athletic performance. This paper proposes a method of rowing sport capture and analysis based on Inertial Measurement Units (IMUs). A canoeist’s motion was collected by multiple miniature inertial sensor nodes. The gradient descent method was used to fuse data and obtain the canoeist’s attitude information after sensor calibration, and then the motions of canoeist’s actions were reconstructed. Stroke quality was performed based on the estimated joint angles. Machine learning algorithm was used as the classification method to divide the stroke cycle into different phases, including propulsion-phase and recovery-phase, a quantitative kinematic analysis was carried out. Experiments conducted in this paper demonstrated that our method possesses the capacity to reveal the similarities and differences between novice and coach, the whole process of canoeist’s motions can be analyzed with satisfactory accuracy validated by videography method. It can provide quantitative data for coaches or athletes, which can be used to improve the skills of rowers.

## 1. Introduction

Rowing, combining wonderful spectacle with heated competition, has become a popular international sport. Sport organizations including professional clubs or nation sport institutions have tried to gain the winning edge through incremental improvement by means of effective and scientific auxiliary training methods for athletes. The behavior of athletes in rowing can be influenced by multiple factors including psychological quality and mentality, physical strength or fitness, technique proficiency level, and so on. Among these factors, the competitive level of athletes plays an important role. In rowing training and competition, the athletes’ competitive level is defined as the standardization and repeatability of stroke action, an efficient and consistent stroke is necessary for achieving a good rowing performance. In the rowing sport of single canoe, the stroke quality, including stroke length, stroke rate, stroke rate variance, propulsion/recovery phase ration, and rhythm is the most vital performance indicator of rower skill proficiency. Stroke quality has been extensively studied by scientist to offer advices for the improvement of athletic performance.

Among the methods being used to analyze the rower’s stroke quality, a video-based method has been adopted in the literature [1,2,3]. Motion detection is limited by the installation conditions of the monitoring device. Only through a specific angle and position to shoot video, will there be occlusion of line of sight and limited shooting angle in the motion. Recent technological development has made miniature inertial devices widely available. McDonnell et al. used inertial sensors attached to the kayak and paddle to capture stroke period and specific signal peak values [4]. Gomes et al. used a 9-degrees-of-freedom IMU mounted on the paddle to analyze the inter-stroke intervals of individual kayakers [5]. However, previous studies mainly focused on measuring stroke quality by mean of equipment, and less attention has been paid to the athlete. Rowing is a coordinated action and involves several muscle groups, it occurs mainly by flexion extension movements, with abduction movement for both limbs, paddling movement only results from the combined effect of the above factors [6,7].

The angles of knees, elbows, waists, and necks are the principal kinematic analysis subjects of individual rower, which were widely studied. Llosa, Mpimis et al. used goniometers to measure the rower’s flexion and extension angles of the elbows [8,9], but it is not suitable to describe rotation motion of the athlete’s limbs and trunk. Said et al. used inclinometers and trigonometric calculations to get the rower’s joint angles variation under simulated conditions [10]. However, the scope of human activities is limited, which is constrained within relatively tight bounds. Wang et al. used IMUs mounted on the canoeist body to capture his motion data, only the stroke phases have been studied [11]. Most studies are limited by the fact that the systematic and quantitative analysis of canoeing sport based on joint movement is relatively not sufficient.

To conduct kinematic analysis on canoeing sport, a method of rowing sport capture and analysis based on IMUs is proposed. For our analysis, the body is considered to be a set of rigid body model, including multiple segments with custom lengths, the adjoining segments are connected by frictionless variable degrees of freedom joints. Singularity-free unit quaternion was used to represent each body segment’s orientation, the joint angles of body parts flexion extension movements were obtained by quaternion operation. The main contributions of this work are as follows.

Use the gradient descent method to fuse the inertial sensor data, obtain the real-time attitude of rower and capture the motion of athletes with different skill levels under realistic conditionsThe effectiveness and accuracy of proposed attitude estimation algorithm have been verified through optical motion capture systemKinematic analysis has been applied to the rowers with different skill levels from statistical perspective, and machine learning algorithm is used to discriminate different proficiency level athletes

The article is structured as follows. Section 2 introduces the hardware and software platform. The experimental methodology is described in Section 3. The algorithm verification results are given in Section 4. The discussion of this study is described in Section 5. Finally, conclusions are given in Section 6.

## 2. System Platform and Data Acquisition

In this article, the motion capturing system is developed by the laboratory of intelligent system of Dalian University of Technology. It consists of several tiny sensor nodes, one transceiver and a set of personal computer (PC) software as shown in Figure 1. Each node contains a MEMS inertial sensor, the device parameters are depicted in Table 1. The new STM32 XL density devices were used as a micro controller chip to receive data from the sensor nodes, and a trans-flash memory card was used to store raw data.

Lora wireless communication is used between slave sensor nodes and the master transceiver. Once the slave nodes receive the start signal from the master unit, they record motion information of rowers and store them into the nonvolatile memory card with the filesystem immediately, the self-made measurement system could be set to high sampling frequency (up to 800 Hz). Figure 2 shows the data-acquisition mode. To validate the proposed algorithm accuracy and check the performance of the self-designed system, simultaneous measurements of joint angle were needed to compare with the high-speed motion camera.

In this study, six participants including two coaches and four novices take part in preliminary studies. They come from the provincial sprint team, and the four novices have more than one year of training experience.They trained for 25 to 30 h a week. They have an average weight of 70 ± 10 kg and height of 1.70 ± 0.10 m. All the participants were fully informed, and consent was obtained. The experimental site was located in the Athletic Training Center, Dalian, Liaoning, China (latitude N 121°25.539′ and longitude E 38°92.963′). During the experiment, miniature sensor nodes were placed on the surface of the canoeist’s body.

## 3. Methods

### 3.1. Overall System Architecture

The athlete’s body is defined as rigid structure based on human anatomy theory, the skeletal structure consists of at most seventeen segments as shown in Figure 3a, and the length of every segment can be defined manually with the height of participants. The nine-degrees-of-freedom inertial sensor nodes are placed on the corresponding limb segment, which is used to obtain the raw acceleration, gyroscope and magnetometer information during data acquisition process. The specific locations of the sensor data sampling points are shown in Figure 3b.

As shown in Figure 4a, the whole system contains three coordinate systems, and each three-dimensional coordinate system is based on standard right-handed 3-D Cartesian coordinate system [12]. The details are as follows:Ground Coordinate System (GCS): It is the navigation coordinate system and complies with the laws of east, north and up (ENU). The Y-axis corresponds to North and the X-axis corresponds to East. This makes the scene an “East North Up” (ENU) coordinate system.Sensor Coordinate System (SCS): It is defined as the coordinate of sensor nodes placed on the body.Body Coordinate System (BCS): The X-axis is perpendicular to the body surface, pointing outward, and the Y and Z axes are orthometric to the X-axis. The BCS is based on the right-hand rule.

The skeletal part of our model has 17 rigid links, including trunk (head, arms, and torso) and thigh, shank, and foot of bilateral lower limbs. Elbow, knee, and ankle were allowed free movement. The joint angle definitions are provided in Figure 4b. In this way, the increase of joint angle corresponds to joint flexion, and vice versa. Rowing movement occurs mainly by joint flexion, we defined the joint angles as shoulder flexion angle (SF), elbow flexion angle (EF), knee flexion angle (KF) and foot flexion angle (FF) [12]. In this paper, we mainly focus on the movement of the upper limbs [13].

At the beginning of the motion capturing process, the magnetometer needs to be calibrated because of soft iron distortion and hard iron distortion in the surrounding environment. Hard-iron distortion originates mostly from permanent magnet and magnetized metal, soft-iron distortion is the result of material that influences, or distorts, a magnetic field-but does not necessarily generate a magnetic field itself, and is therefore not additive. Ellipsoid fitting method is adopted in this paper to eliminate ferromagnetic interference, and the soft iron is relatively small and is negligible [14]. The ellipsoidal fitting results are shown in Figure 5.

At the end of sensor signal preprocessing, gradient descent method was used to fuse multi-sensor data. The pelvis was set as a reference point, each segment attitude could be calculated through multiple iterative loop from the initial state based on quaternion rotation and translation. The joint angles (foot, knee, shoulder, elbow) were computed from the elevation angle of adjacent segments. Then we can analyze the canoeist stroke quality under different skill proficiency level and improve their athletic performance via quantitative analysis. A more detailed description of the overall algorithmic steps and their implementation is given in subsequent sections. Figure 6 shows the schematic overview of the proposed method. When only upper or lower limb activities are discussed, the body model and iteration operation can be simplified and it is feasible to merely consider active segments parts.

### 3.2. Motion State Update Based on Quaternion Method

To avoid gimbal lock, quaternion is used to describe the body segment orientation in this paper, it has the following form as shown below, where i, j, and k are the standard orthonormal basis represented by unit vectors in 3D space.
(1)$$q=\left(q_{0},q_{1}i,q_{2}j,q_{3}k\right).$$

In the initial stage, the canoeists were required to stand with arms down for set interval time, the action duration depended on time-series length used in the initial stage, so the coordinate system BCS is overlapped with the coordinate system GCS. The initial quaternion rotation from SCS to BCS is similar to the quaternion from SCS to GCS. That is, $q_{S,init}^{B}\approx q_{S,init}^{G}$. The $q_{S,init}^{G}$ can be obtained by magnetometer and accelerometer measurement value according to [15]. Because the sensors were tied on the body surface in a fixed position, $q_{S}^{B}$ is equal to approximately $q_{S,init}^{B}$. The initial quaternion $q_{B,init}^{G}$ can be obtained from the following formula, where ∗ denotes the conjugate matrix.
(2)$$q_{B,init}^{G}=q_{S,init}^{G}⊗{\left(q_{S}^{B}\right)}^{*}=q_{S,init}^{G}⊗q_{B}^{S}.$$

During the process of motion capture, if the quaternion of sensor node in GCS is known, the rotation of each limb segment at any given instant could be obtained from the previous time point based on $q_{B}^{G}=q_{S}^{G}⊗q_{B}^{S}$ iteration. In the next step, the reference point is defined at the pelvis, and the length of each segment is predefined according to the participants, so the attitude of each segment in initial state can be obtained through the iteration of the relationship skeletal segment.

The gradient descent algorithm is adopted to update the value of $q_{S}^{G}$ based on quaternion in this paper [16]. The $q_{S}^{G}$ is obtain from Equation (Equation 3).
(3)$$q_{S}^{G}\left(t\right)={\hat{q}}_{S}^{G}\left(t-1\right)-\xi \frac{\nabla f({\hat{q}}_{S}^{G},{\hat{p}}^{G},{\hat{p}}^{S})}{∥\nabla f({\hat{q}}_{S}^{G},{\hat{p}}^{G},{\hat{p}}^{S})∥}.$$

In this formula, ${\hat{p}}^{G}=\left[0,p_{x}^{G},p_{y}^{G},p_{z}^{G}\right]$ is the measured direction of the field in the sensor frame, ${\hat{p}}^{S}=\left[0,p_{x}^{S},p_{y}^{S},p_{z}^{S}\right]$ is the rotation quaternion, the object function f(x) between the ${\hat{p}}^{G}$ in GCS and ${\hat{p}}^{S}$ in SCS is as follows.
(4)$$f\left({\hat{q}}_{S}^{G},{\hat{p}}^{G},{\hat{p}}^{S}\right)={\left({\hat{q}}_{S}^{G}\right)}^{*}⊗{\hat{p}}^{G}⊗{\hat{q}}_{S}^{G}-{\hat{p}}^{S}.$$

Therefore, the gradient of object function is calculated by Equation (Equation 5), and the corresponding Jacobian matrix can be obtained.
(5)$$\nabla f\left({\hat{q}}_{S}^{G},{\hat{p}}^{G},{\hat{p}}^{S}\right)=J^{T}\left({\hat{q}}_{S}^{G},{\hat{p}}^{G}\right)f\left({\hat{q}}_{S}^{G},{\hat{p}}^{G},{\hat{p}}^{S}\right).$$

When all the segments posture of the rigid body model were obtained from the relative skeletal segment iteration calculation, the vector angle, i.e., the joint angle could also be solved by inverse cosine between two adjacent skeletal segment vectors.

### 3.3. Experimental Setting between Self-Made and Standard Systems

To verify the reliability of the self-made inertial motion capture system, we compare data from our developed system to standard optical motion capture system. Consider the environmental factors, the contrast experiment was conducted in the indoor scene. The subjects were instructed to wear specific clothing, and the reflective marker and sensor nodes were all placed on the upper limbs. The motion capture processes between self-made and commercial optical system were initiated simultaneously. The optical instrument was treated as golden standard device produced by the Natural Point Company in the United States. The system consists of 12 cameras, 25 makers, and the motion capture software which is called Motive. The 12-camera motion capture system tracked 25-retro-reflective markers placed on the subject’s pelvis, right and left arms, shoulder, and torso. The markers trajectories were measured at 360 Hz during a static trial and movement at a self-selected speed. The arrangement of the field experiment is illustrated in Figure 7.

After the completion of the contrast test, each participant was required to perform the standardized rowing movement for 200 meters race, the inertial data collection was performed in synchronization with the video recording, a high-speed camera (Sony FDR-X3000R) at a sampling rate of 200 Hz was used to track the motion of the canoeist, and video analysis was conducted using open source software Kinova (version 0.8.22). Because action camera is working with a limited frame rate, the systematic error is inevitable, but it is with acceptable limits, so the video samples were served as the benchmark for labeling the inertial joint angle time series.

## 4. Results

To evaluate the performance of proposed method based on inertial sensor-based motion capture system, the complete protocol consists of the following steps: (1) The accuracy of our self-made motion capture system is verified by means of comparison with the standard optical system; (2) Stroke quality analysis between novice and coach based on joint angles under real water sports conditions. (3) Machine learning algorithms are conducted in the division of different proficiency level athletes.

### 4.1. Performance Comparison between Self-Made and Standard Systems

During the experiment, the participant was required to move in the visual areas of optical motion capture system, with extension of both upper arms. The coordinate system between optical and inertial capture system is not coincided, so the raw motion data need to be transformed for comparison.

Figure 8 shows the contrast graph of flexion extension angles versus the same joint angle deduced from optical cameras measurements. The specific contents of joint movement included shoulder and elbow joint on both body sides are shown in Figure 4. We defined them as left shoulder flexion (SFl), right shoulder flexion (SFr), left elbow flexion (EFl) and right elbow flexion (EFr) respectively, experimental data associated with them are represented by scattered plot and were fitted by straight lines [17]. The linear fitting slopes of four sets of motion data were 0.910, 0.971, 0.971, and 1.043, respectively. The respective corresponding correlation coefficient were 0.995, 0.990, 0.995, and 0.996, respectively. Bland-Altman analysis is shown in Figure 9. The optical system measurement values were used as the standard reference, and Table 2 summarizes the relative error on the results obtained from self-developed inertial sensor-based capture system. The result illustrates that our developed devices are reliable and measurement errors are well controlled.

### 4.2. Motion Restruction Based on Proposed Method

The common definition of complete stroke behavior is based on the contact area of the paddle blade relative to the water, a total of four critical positions were chosen and used for separating the stroke phase including catch, immersion, extraction and release [3]. Catch occurred at the first contact between the paddle blade and water. When the paddle blade was fully submerged, it was defined as immersion. When the blade was just emerging from the surface of the water, it was defined as extraction, and release was the last contact between the blade and the surface of the water. The entry, pull, exit, and aerial are the sub-phases, and the first three phases were combined into propulsion phase. The details of the motion phase sequence definition are shown in Figure 10. The athlete motions were recorded at 360 Hz with a sagittal-view video camera from around 10 meters during 200 meters time trials. As can be seen, the motion of canoeist can be tracked and reproduced. Due to space constraints, the motion of canoeists is mainly upper limb movement. Therefore, the flexion-extension of the elbow and shoulder is the key part to reflect the athletic performance, and the variations of SFl, SFr, EFl and EFr are the emphases of our research.

### 4.3. Stroke Quality Analysis Based on Proposed Method

The most commonly used evaluation criteria of stroke quality in rowing sport are stroke rate (cadence), stroke length, stroke variance, propulsion/recovery phase ratio and stroke force. The four curves of two joints of coach and novice are shown in Figure 11 and Figure 12. From the two graphs, the evaluation parameters of stroke quality can be derived. The top wide blue and red lines are duration of each stroke cycle, which were analyzed by manual annotation using an action camera. The middle wide blue and red lines are the signal periods, which are easily deduced by peak-pick algorithm. Obviously, owing to the inevitable systematic errors in visual method (200 Hz frame rate), the latter performs much more accurate than the former method. The stroke rate (cadence) can also be calculated from the reciprocal of signal period, which was the most frequently extracted metric of rowing performance. The stroke variance can be obtained from the signal period fluctuation. The value of the coach’s stroke cycle period is 1.72 ± 0.05 s. The value of novice’s stroke cycle period is 1.71 ± 0.08 s. The durations of the stroke cycle in 200 meters trip recorded for the coach and novice are presented in Figure 13. It can be seen from the graph that the stroke variability of the coach is relatively stable.

The propulsion/recovery rate is generally used to describe an athlete’s rhythm, which is the most important factor for athletes [18,19]. Deficiencies in the rhythm of rowers significantly decrease the velocity of the canoe. The performance of canoe increasing the propulsion duration while decrease the recovery duration in each stroke cycle. According to [11], we take the following steps to obtain the duration of propulsion and recovery phase. First, The sliding window was used to divide the time series of four joint curves (SFl, SFr, EFl, and EFr) into trials (time series segments), the length of each segments is ten sampling points, and the overlapped field length of sliding window is five sampling points. After window segmentation, video record was used to estimate whether it belong to the propulsion state or recovery state, and determine the true label of each segment; Second, feature extraction was applied to each segmentation record, standard statistics, time-domain and frequency-domain (or spectral-domain) based features were extracted on per overlapping 25 milliseconds windows [20]. After feature extraction, the feature matrix was formed, and each row represented one unique combination of features; Finally, the support vector machine (SVM) was used as the classification algorithm in this paper. The labeled training samples were used as training set, grid-search method was used to find the optimal model parameter. After training, the classification model was obtained, and the remaining samples were used to characterize the accuracy of the selected model. Results of prediction from the trained model are shown in Figure 14. The propulsion/recovery rate of the coach is 1.98 ± 0.26, the propulsion/recovery rate of novice is 2.05 ± 0.51. The predicted results of the proposed method were comparable with the video capture-based method.

### 4.4. Statistical Analysis of Canoeing Procedure

To further analyze the athlete’s motion characteristics of different skill proficiency level, statistical analysis was performed for both sides upper limbs in both novice and coach groups [21]. Because the body parts accomplished the action were reverse, the curves of joint angle for comparison need to be adjusted, that is the novice’s EFl versus coach’s EFr, novice’s EFr versus coach’s EFl and so on, the details are shown in Figure 15 and Figure 16.

The standard methods recommended for the statistical analysis were used in the present study [22,23,24], the statistical meaning of these parameters is as follows: ROM: range of motion; MAX: maximum value; MIN: minimum value; MEAN: mean value; SD: Standard deviation. To provide an intuitive understanding of the difference between different proficiency level participants. Figure 15 and Figure 16 illustrate the curves of joint angle of elbow and shoulder during a stroke cycle. In these graphs, the red solid lines represent the mean of group time recorded; the black dashed lines represent the maximum and minimum mean values; the light red shaded area indicates the ROM between MAX and Min. In Figure 15 and Figure 16, each stroke was divided into four phases according to Figure 10.

Table 3 shows the calculation outcomes of 372 records, the results were combined with Figure 15 and Figure 16, which produce the following conclusions: Under the premise that participants were instructed to perform as normally and accurately as possible, when we compare data from novice versus coach, it can be found that the standard deviation of the elbow was generally higher than the shoulder. This is because the forearm contacts close to the paddle blade [25]. The contact between the water surface and blade affects wrist movement, which in turn affects forearm and upper arms. When we compare the novice’s EFr and coach’s EFl, the standard deviation of coach is slight smaller than novice’s, it also indicated that the pattern of coach action was more consistent than novice, and with a stable performance. From Table 3, it can be seen that the ROM of the coach is roughly equivalent to novice’s, whether elbow flexion extension or shoulder’s. However, this was not true for the other parameters, the coach’s elbow flexions were higher than novice’s. As for the shoulder joint, the converse was true. These results showed that the upper arm and shoulder were used by novice to complete the rowing action, and it is not suitable for keeping balance, hence the boat speed was affected [26].

### 4.5. Athlete Recognition of Different Proficiency Level

Sport behavior has always been one of the hot topics in the wearable device application field. To explore the representative athlete’s characteristics of different proficiency level, machine learning algorithms were used to classify coach and novice based on features matrix of four joint angles dataset, and find out the salient features to distinguish coaches from novices.

A total of 33 standard time-domain and frequency-domain features are listed in Table 4. Feature extraction was employed upon each record of four joint angles, including SFr, SFl, EFr, and EFl. The length of each record was determined by the peak-to-peak value of the should flexion-extension curves, which were depicted in Figure 11 and Figure 12. In total, 132 features were extracted for each record. Furthermore, principal component analysis (PCA) was carried out on the feature dataset, Figure 17a demonstrated the two-dimensional representation of conservation features. Most of the variance between records (64.21%) was explained by component1, The overall rate of contribution of the first two primary components is 87.28%. It shows that the coach group can be separated from the novice group based on joint angle-based features.

At first, all of the 132 features were used while training the model. To reduce computational costs and storage requirements, and get a simpler model that is less likely to overfit. Feature selection is adopted to remove features that are redundant or do not carry useful information. It can reduce the size of the model and can be readily applied. Neighborhood component analysis (NCA) is a non-parametric and embedded method for selecting features with the goal of maximizing prediction accuracy of classification algorithms [27]. The relationship between weight and feature index is depicted in Figure 17b. When the feature selection is finished, the 6 features that the weight >0.1 is remained, they are all the autocorrelation features of four joint angles. The results are basically consistent with the previous results [28], and the empirical analysis results in this paper are valid.

The feature dataset is randomly divided into two independent sets. The 75% of the dataset is selected for training classification model. The 25% of remaining dataset is used for testing model. During the training process, a random 10% data of training dataset was used as validation dataset, the modes was guided by observing the cross-validation accuracy during training and choosing new parameters until no further improvement could be made. This separation was performed at the participant level. This means that all feature dataset from a athlete was included in the same person (training dataset, validation dataset, and testing dataset). All these measures ensured testing dataset contained only information that had not been encountered by the model during training. The four types of machine learning algorithms, including Support Vector Machine (SVM), Logistic Regression, Decision Tree and XGBoost are conducted on the feature dataset for classifications. The grid search method is used to find the optimal parameters of each algorithm. The receiver operating characteristic curve (ROC) could give more informative metric to check the quality of classifiers. The quality of the multiple model was evaluated through its measures of sensitivity and specificity with the establishment of a ROC curve [29]. The area under ROC curve (AUC) was always used to test for sensitivity and specificity of each algorithm. The classify ability of different algorithm comparison are shown in Table 5. The hyperparameters are exploited by a k-fold cross validation procedure for the experiments. All classification problems were performed using Windows 10 LTSC, running python 3.6, and using Scikit-learn library version 0.21.3. It can be found that the overall recognition is satisfactory when four joint angles were dealt with. The XGBoost algorithm reaches the highest recognition accuracy that is 100%, and the performance of SVM algorithm is slightly worse. The recognition accuracy of XGBoost is 98.51% when using the selected features. It can be seen that the method using joint angle based on IMU motion capture technology has advantages in accuracy of the rowers’ proficiency level recognition.

## 5. Discussion

The wearable inertial sensor network has widely been adopted as a training assistant manner to give useful feedback for coaches during practice, and it can provide quantitative insight into each aspect of rowing activities. The information fusion of multi-sensor can produce insightful metrics, to address this issue, this paper proposes an innovative approach based on data fusion technology to estimate the motion posture of rowers, and provides the detailed kinematic analysis of joint flexion extension from different proficiency level.

The developed system can track the rower’s action accurately compared with the optical motion capture system, and the gradient descent method was used to eliminate the rotation error from sensor coordinate system to navigation coordinate system, and update the real-time attitude of the experimenters. The implementation of rowing motion capture can provide not only stoke quality analysis, but also additional statistical information, more insightful metrics can be obtained by the advanced sensor fusion algorithm, the waveform parameters (MEAN, ROM, MAX, MIN) of joint angles provide a detailed description of the similarities and differences between novice and coach compared with the literature [10]. On the other hand, the sampling rate of inertial system can be set at a higher rate (800 Hz), and it reflects an ability to obtain more actionable information compared with the video movement analysis [3]. Furthermore, multiple machine-learning algorithms were used to distinguish novice from experienced rowers, and satisfactory results have been achieved. In addition, it can tell the novice rower what their exact deficiency in technique is [30].

It should be noted that although the inertial sensor system has advantages of being portable and without space limitations, athletes would feel discomfort after wearing the sensor nodes for more than half an hour [31]. In this case, there is a great need for more comfortable motion-monitoring solution, or fewer miniature sensor nodes were used on the premise that the performance is guaranteed. In addition, video recording was used as a means of determining the true labels of propulsion and recovery phase, the systematic error (e.g., frame rate) was unavoidable, and to accurately determine the touchpoint between paddle blade and water is difficult, therefore, this might lead to inaccurate phase divisions, and it might have influenced the results of our secondary predicted outcomes.

## 6. Conclusions

In this work, we introduced a rower motion capture and analysis system using an inertial sensor network. The field water sport experiments validated the comprehensive and detailed information that can be obtained from the proposed system. During the development process, a free rigid segment model was proposed and the attitude of each body segment could be obtained by the iteration calculation from the pelvis rotations. In addition, the selection of body segments can be tailored to the application. For practical applications, we demonstrated that our method is able to achieve the comparable accuracy to the standard optical motion capture system.

In future work, we plan to extend our work as follows: a more detailed profile of the sub-phase, including entry, pull, exit and aerial could be studied based on joint angle, and it could promote the effective use of systematic observation strategies for coaches. In the sub-phase, legs performed driving tasks, and deficiencies in leg movement would significantly influence performance of the boat, these factors would be considered in the future. In addition, as the number of sensors bound to the human body is excessive, which made the rowers feel uncomfortable We are currently developing a lightweight and miniature wearable network module, and it can be integrated into electronic products, such as wristband. A more comprehensive water sport athlete monitoring system will be established in the future.

## Figures and Tables

**Figure 1 sensors-20-02110-f001:**
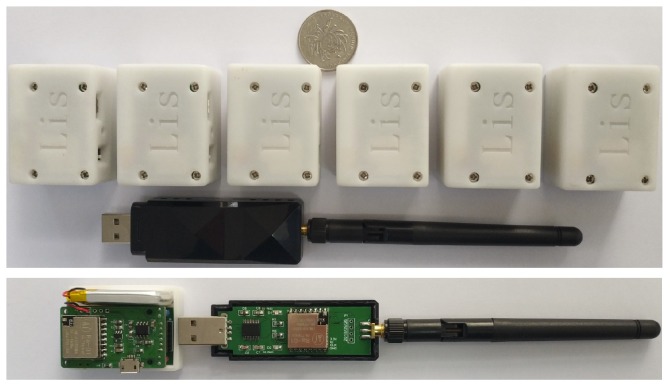
The device appearance and its disassembled prototype of the self-made inertial sensor-based motion capture.

**Figure 2 sensors-20-02110-f002:**
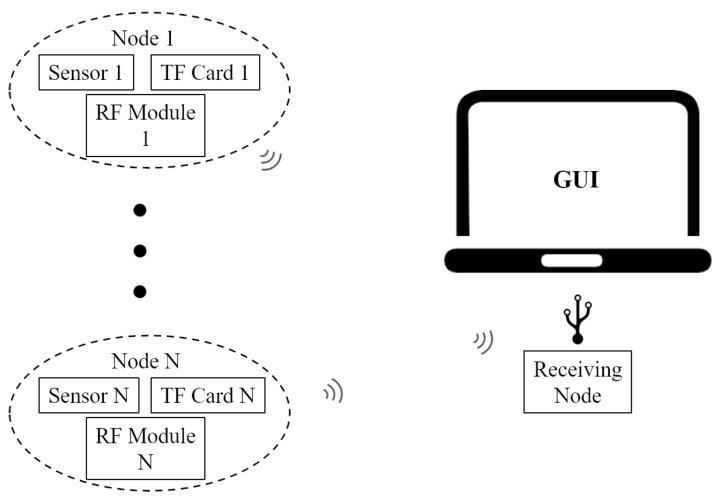
The block diagram of the experimental system.

**Figure 3 sensors-20-02110-f003:**
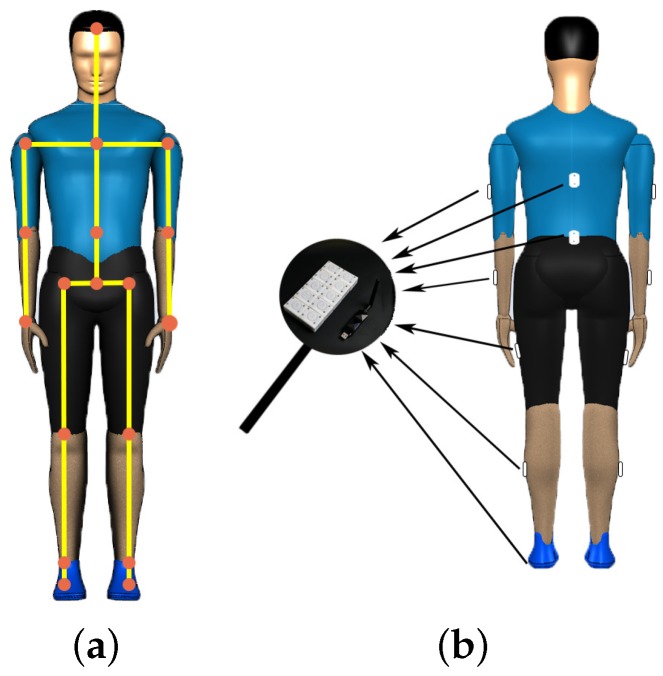
The human rigid structure model and the device locations. (**a**) the whole body structure definition with rigid body model. (**b**) postions of sensor nodes during the experiment.

**Figure 4 sensors-20-02110-f004:**
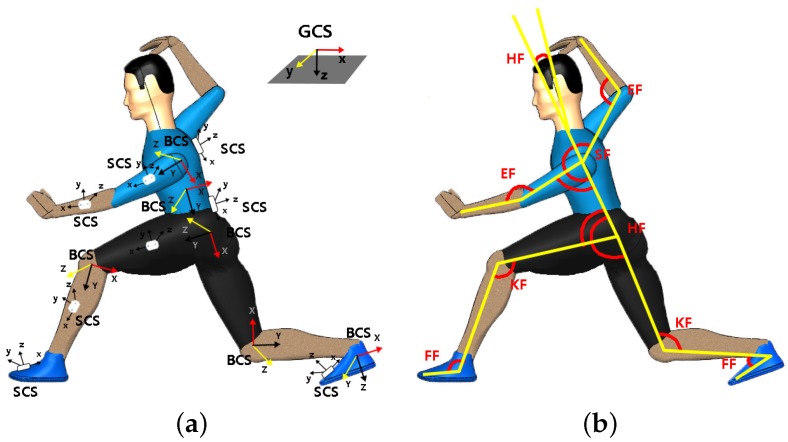
The definition of coordinate system and the limb flexion joint angle. (**a**) the three coordinate systems in self-made inertial motion tracking system. (**b**) the joint angle definition for the athlete model.

**Figure 5 sensors-20-02110-f005:**
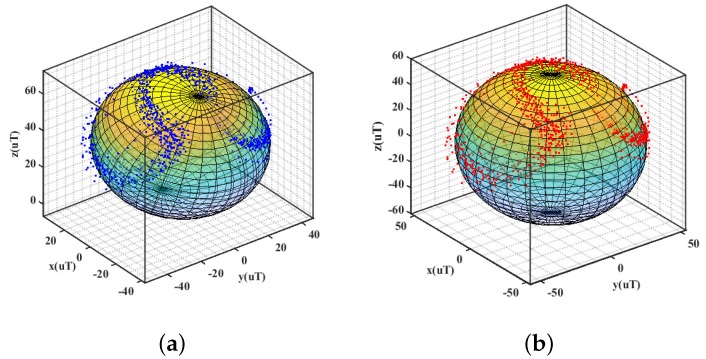
The calibration outcome of magnetometer. (**a**) before fitting. (**b**) after fitting.

**Figure 6 sensors-20-02110-f006:**
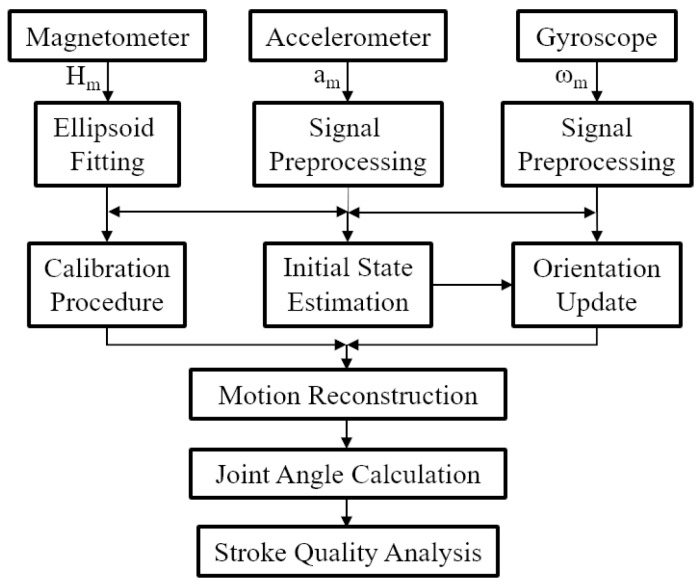
The schematic overview of the proposed inertial motion capture method.

**Figure 7 sensors-20-02110-f007:**
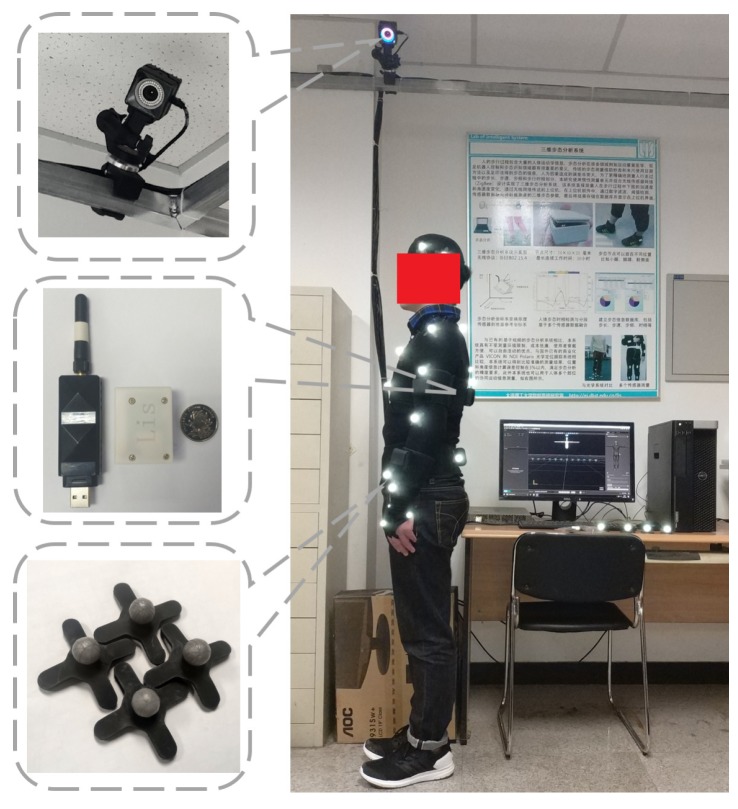
The overview of contrast tests between self-made and standard optical capture system.

**Figure 8 sensors-20-02110-f008:**
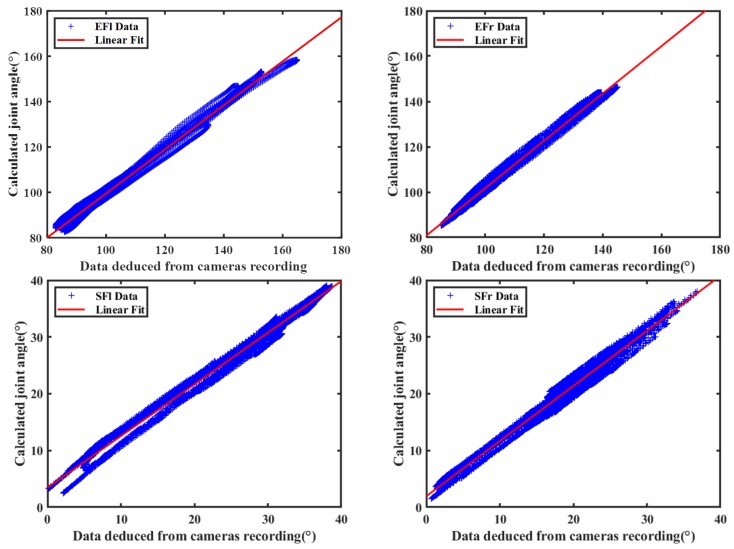
Contrast result of elbow and shoulder flexion-extension angle.

**Figure 9 sensors-20-02110-f009:**
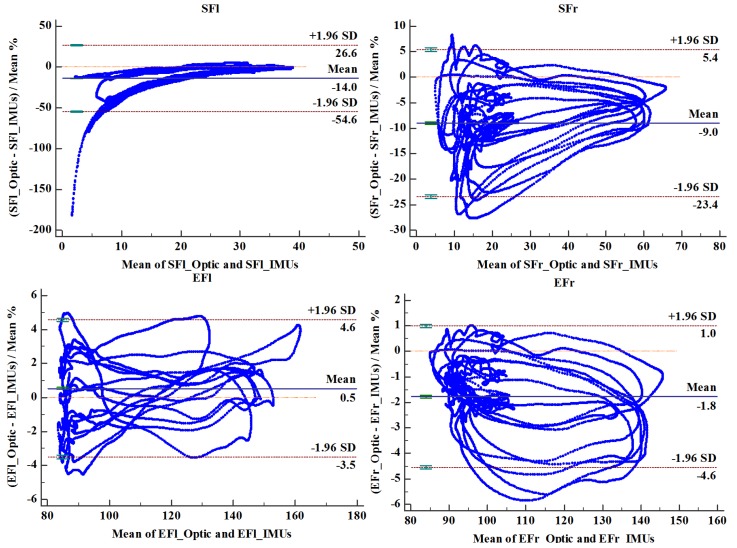
Bland-Altman analysis plot for upper limbs joint angles.

**Figure 10 sensors-20-02110-f010:**
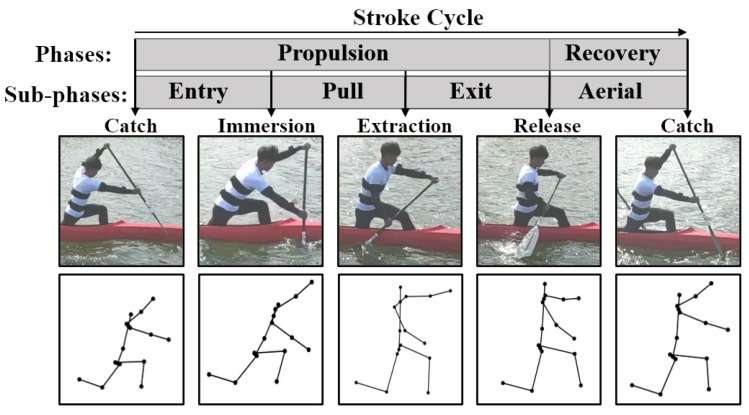
Phases definition of the canoeing stroke cycle (entry, pull, exit and aerial) separated by paddle positions (catch, immersion, extraction and release)

**Figure 11 sensors-20-02110-f011:**
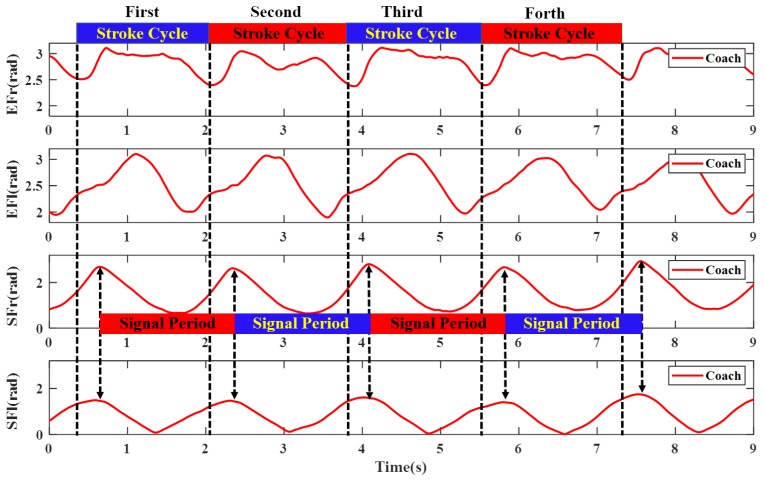
The elbow and shoulder flexion joint angle of coach.

**Figure 12 sensors-20-02110-f012:**
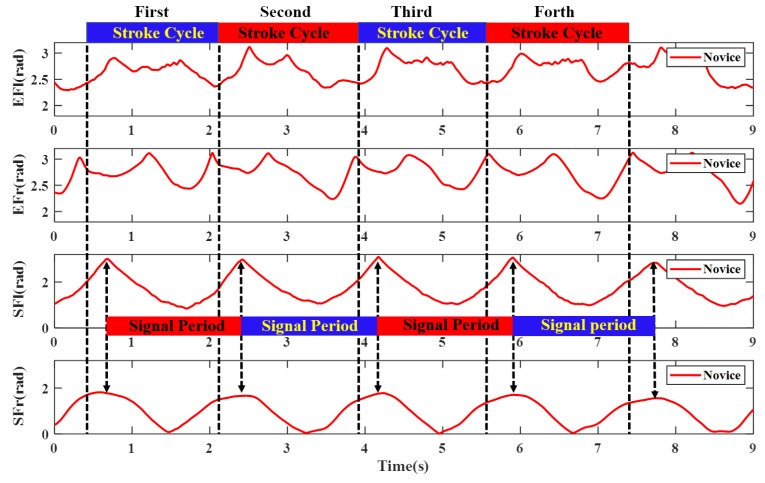
The elbow and shoulder flexion joint angle of novice.

**Figure 13 sensors-20-02110-f013:**
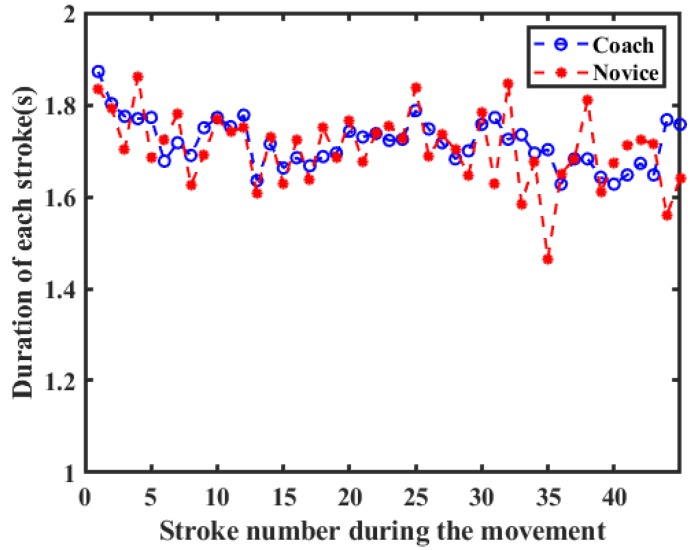
The duration of each stroke versus the stroke numbes during the rowing movement.

**Figure 14 sensors-20-02110-f014:**
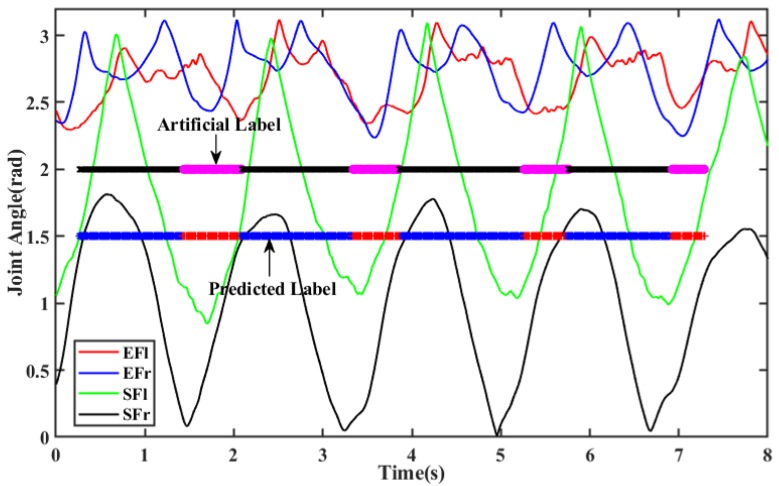
Schematic diagram of propulsion/recovery phase predicted based on joint angle.

**Figure 15 sensors-20-02110-f015:**
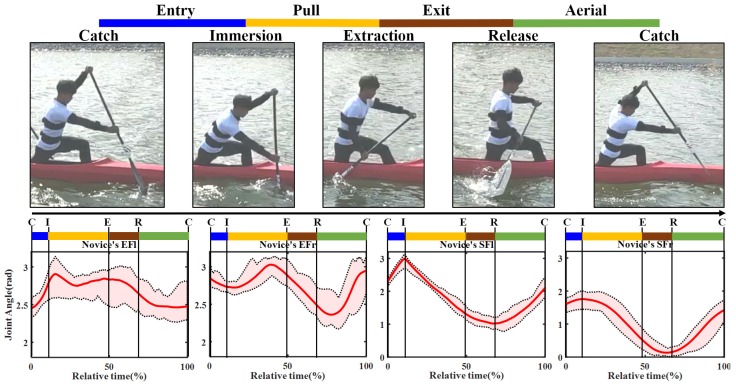
The joint flexion angle extension variation of novice on both sides of the body.

**Figure 16 sensors-20-02110-f016:**
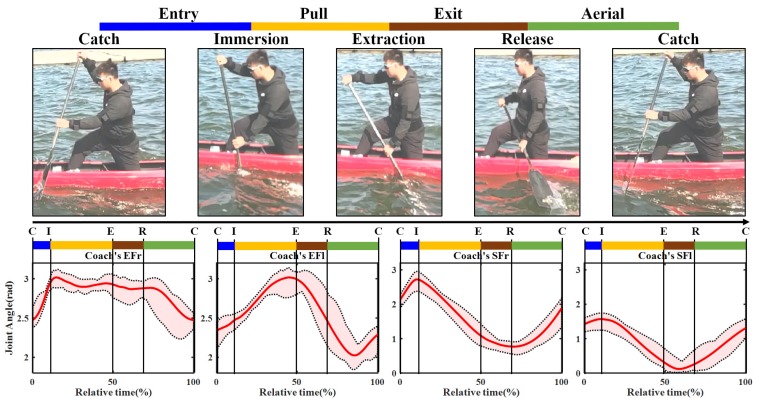
The joint flexion angle extension variation of coach on both sides of the body.

**Figure 17 sensors-20-02110-f017:**
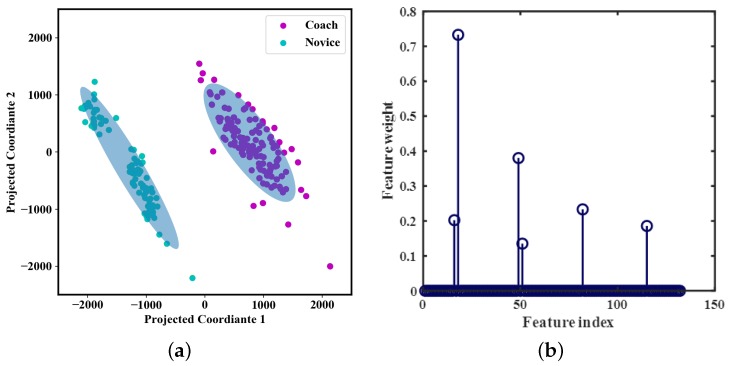
Principal component analysis (PCA) results and feature selection results. (**a**) the scatter plot of principal component analysis. (**b**) a plot of feature selection by Neighborhood component analysis (NCA).

**Table 1 sensors-20-02110-t001:** The device parameters of the sensor node.

Unit	Accelerometer	Gyroscope	Magnetometer
Dimensions	3 axis	3 axis	3 axis
Sensitivity (/LSB)	0.833 mg	0.04 deg/s	142.9 uguass
Dynamic Range	±18 g	±1200 deg/s	±1.9 gauss
−3 dB Bandwidth (Hz)	330	330	25
Nonlinearity (%FS)	0.2	±0.1	0.1
Misalignment (deg)	0.2	0.05	0.25

**Table 2 sensors-20-02110-t002:** Joint angle measurement error.

Joing Angle	Mean Error (%)	SD (%)
SFl	3.72	1.88
SFr	2.19	1.23
EFl	1.20	1.02
EFr	2.37	1.15

**Table 3 sensors-20-02110-t003:** Evaluation of joint angle parameter.

Mean ± SD		Elbow Flexion Angle (rad)				Shoulder Flexion Angle (rad)			
		ROM	MAX	MIN	MEAN	ROM	MAX	MIN	MEAN
Coach:	Right Side	**0.61 ± 0.09**	**3.06 ± 0.06**	**2.45 ± 0.08**	**2.83 ± 0.05**	**2.05 ± 0.11**	**2.81 ± 0.08**	**0.75 ± 0.09**	**1.57 ± 0.07**
	Left Side	1.06 ± 0.11	3.04 ± 0.07	1.97 ± 0.07	2.55 ± 0.03	1.59 ± 0.08	1.64 ± 0.07	0.04 ± 0.03	0.85 ± 0.05
Novice:	Right Side	0.81 ± 0.11	3.11 ± 0.22	2.30 ± 0.11	2.74 ± 0.03	1.75 ± 0.08	1.82 ± 0.09	0.07 ± 0.05	0.98 ± 0.06
	Left Side	**0.62 ± 0.13**	**3.00 ± 0.08**	**2.39 ± 0.08**	**2.68 ± 0.05**	**2.02 ± 0.11**	**3.02 ± 0.06**	**1.00 ± 0.08**	**1.79 ± 0.04**

**Table 4 sensors-20-02110-t004:** List of feature vectors.

Feature Name	Description	Number
mean	Mean value	1
median	Median value	1
std	Standard deviation	1
mad	Median absolute value	1
quantile (1–2)	Signal percentile	2
iqr	Inter quartile range	1
skewness	Time signal skewness	1
kurtosis	Time signal kurtosis	1
var	Time signal variance	1
sigentropy	Signal entropy value	1
sepentropy	Spectal entropy of the signal	1
powersp (1–3)	Power spectrum features	3
acorr (1–3)	Autocorrelation features	3
spwf (1–15)	Spectral power features	15

**Table 5 sensors-20-02110-t005:** Classification performance and the optimal hyperpapameters of the proposed algorithms.

	Before Feature Selection			After Feature Selection		
	Accuracy	AUC	Hyperparameter	Accuracy	AUC	Hyperparameter
SVM	100%	1.00	C: 1, gamma: 0.01 kernel: rbf	96.82%	0.97	C: 2000, gamma: 0.001 kernel: rbf
Logistic Regression	98.51%	0.98	C: 10, multi_class: multinomial penalty: l2, solver: lbfgs	95.52%	0.95	C: 50, multi_class: ovr penalty: l2, solver: lbfgs
Decision Tree	94.28%	0.93	criterion: gini, max_depth: 8	94.02%	0.92	criterion: gini, max_depth: 7
XGBoost	100%	1.00	n_estimators: 30 max_depth: 3 learning_rate: 0.25 subsample: 0.8 colsample_bytree: 0.5 min_child_weight:1	98.51%	0.99	n_estimators: 35 max_depth: 2 learning_rate: 0.25 subsample: 0.7 colsample_bytree: 0.5 min_child_weight: 3

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
