# Peer review of "Canoeing Motion Tracking and Analysis via Multi-Sensors Fusion"

_sensors, 2020, doi:10.3390/s20072110_

Round 1

Reviewer 1 Report

General comments:

The authors present an interesting work on motion tracking in canoeing using wearable sensor data. Athlete performance objective measures using wearable technology is an important topic. This work presents as very interesting for the readers of Sensors but the manuscript requires extensive English grammar review and the structure of this manuscript should be also revised before considering for publication. Further details of the methodologies should also be reported. All the figures’ captions should be expanded to explain better the contents of each figure. A more detailed assessment is presented below.

Specific comments:

  • Page 3, lines 87-93: Was there any ethical approval for this study? Did the participants sign an informed consent form?
  • Page 4, Fig. 3: This figure is not needed as it does not present any critical information.
  • Page 4, line 99: Where exactly were the sensors placed?
  • Page 5, line 117: Please explain briefly what a soft and hard iron distortion is?
  • Page 5, Fig. 5: Please explain in more detail this figure.
  • Page 6, line 133: Please state for how long the arms were down.
  • Page 6, Eq. (2): Please explain the meaning of “*” in this equation.
  • Page 7, Line 163: All the explanations regarding equipment and experimental protocols should go to the Methods sections. In the Results section, only the results should be described. Also, please explain in detail the motion capture system (how many cameras, how many markers, location of the markers).
  • Page 7, lines 174-176: Please add also Bland-Altman plots.
  • Page 8, Fig. 9: This figure should be enlarged (perhaps the format of a 2x2 panel would be a better arrangement).
  • Page 8, lines 180-199: This sub-section should be moved to the Methods sections. Also, please explain how were the cameras mounted (or where).
  • Page 10, line 226: Explain the reason for choosing SVM instead of another classifier.
  • Page 10, lines 231-232: What do you mean with “predicted results were basically the same for all the ground truth”?
  • Page 11, line 239-241: Please explain the statement: “recommended”, by who recommended? Why these statistical measures?
  • Page 12, line 278: Please explain what do you mean with the “embedded” method? Also, which software was used to run these feature selection and the classification methods?
  • Page 13, Fig. 17: Please add labels to the groups of the left panel.
  • Page 13, line 292: It is confusing if you used cross-validation or training and testing sets as stated in the previous paragraph (page 12, lines 283-284). Please clarify.
  • Page 13-14: Discussion: please compare your methods and results with existing literature.

Reviewer 2 Report

This manuscript described a way to quantify athletes' performance during canoeing using inertial measurement units. I have some concerns regarding the manuscript.

First the goal of this study is not very clear to me. Why is it necessary to quantify athletes' performance? Why is IMU a good candidate in terms of method? Particularly in the introduction, the major part of the introduction needs to be re-written because the current for does not really explain why this study needed to be done.

Also, importantly, did the authors get IRB approval before data collection (this could be an important ethical issue).

Reviewer 3 Report

Dear Authors,

Please provide better introductory part, which will reflect the state of the art in motion tracking and accelerometer sonsors used for sportsmans monitoring, as now it is excessively narrowed to oaring sports only.

Sincerely,

Reviewer

Round 2

Reviewer 1 Report

The authors answered most of my concerns, but their changes are difficult to track on the revised manuscript. For example, I cannot find where the statements about ethical approval have been incorporated in the text. Please add Page and Lines where the changes were made in the manuscript for each comment of in my previous review.

The Bland-Altman plots were not incorporated in the revised manuscript, please also use only dots in these plots and remove the connecting lines. 

Reviewer 2 Report

I still don't see sentences regarding IRB approval and obtaining written informed from every single participant. Please make sure the authors obtained such documents from every participant.
